# Efficacy and Safety of Carbon-Ion Radiotherapy for Stage I Non-Small Cell Lung Cancer with Coexisting Interstitial Lung Disease

**DOI:** 10.3390/cancers13164204

**Published:** 2021-08-20

**Authors:** Naoko Okano, Nobuteru Kubo, Koichi Yamaguchi, Shunichi Kouno, Yuhei Miyasaka, Tatsuji Mizukami, Katsuyuki Shirai, Jun-ichi Saitoh, Takeshi Ebara, Hidemasa Kawamura, Toshitaka Maeno, Tatsuya Ohno

**Affiliations:** 1Gunma University Heavy Ion Medical Center, 3-39-15 Showa-machi, Maebashi 371-8511, Japan; kubo@gunma-u.ac.jp (N.K.); y.miyasaka@gunma-u.ac.jp (Y.M.); kawa@gunma-u.ac.jp (H.K.); tohno@gunma-u.ac.jp (T.O.); 2Department of Radiation Oncology, Gunma University Graduate School of Medicine, 3-39-15 Showa-machi, Maebashi 371-8511, Japan; 3Division of Allergy and Respiratory Medicine, Integrative Center of Internal Medicine, Gunma University Graduate School of Medicine, 3-39-15 Showa-machi, Maebashi 371-8511, Japan; ckpnt341@yahoo.co.jp (K.Y.); contra.since2005@gmail.com (S.K.); mutoyu03@gunma-u.ac.jp (T.M.); 4Department of Radiology, Jichi Medical University, 3311-1 Yakushiji, Shimotsuke 329-0498, Japan; tmizukam36@gmail.com (T.M.); junsaito@med.u-toyama.ac.jp (J.-i.S.); 5Division of Radiation Oncology, Department of Radiology, Faculty of Medicine, Academic Assembly, University of Toyama, 2630 Sugitani, Toyama 930-0194, Japan; kshirai@jichi.ac.jp; 6Department of Radiation Oncology, Kyorin University, 6-20-2 Shinkawa, Mitaka 181-8611, Japan; tebara@ks.kyorin-u.ac.jp

**Keywords:** non-small cell lung cancer, stage I, carbon-ion radiotherapy, interstitial lung disease

## Abstract

**Simple Summary:**

Interstitial lung disease (ILD) is a risk factor for lung cancer, but the treatment options are often limited because of concerns that ILD may worsen with treatment. In this study, we analyzed whether the presence or absence of ILD affects the outcome of carbon-ion radiotherapy (CIRT) for clinical stage I non-small cell lung cancer (NSCLC). For all cases, CT and clinical data were reviewed by a respiratory physician to determine the presence of ILD. Overall survival and disease-specific survival were lower in patients with ILD than in patients without ILD. There was no significant difference between the ILD group and the non-ILD group with respect to safety. CIRT was not associated with significantly more side-effects in patients with ILD than in patients without ILD. Coexisting ILD was a poor prognostic factor with respect to CIRT for clinical stage I lung cancer, as reported for other treatment methods.

**Abstract:**

Interstitial lung disease (ILD) is a risk factor both for the development and treatment failure of lung cancer. In this retrospective study, we analyzed the outcome of carbon-ion radiotherapy (CIRT) in 124 patients with clinical stage I non-small cell lung cancer (NSCLC), of whom 26 (21%) had radiological signs of pre-existing ILD. ILD was diagnosed retrospectively by a pulmonologist based on critical review of CT-scans. Ninety-eight patients were assigned to the non-ILD group and 26 patients (21.0%) to the ILD group. There were significant differences in pre-treatment KL-6 values between the two groups. The three year overall survival and cause-specific survival rates were 83.2% and 90.7%, respectively, in the non-ILD group, and 59.7% and 59.7%, respectively, in the ILD group (between-group differences, *p* = 0.002 and *p* < 0.001). Radiation pneumonitis worse than Grade 2 was observed in three patients (3.0%) in the non-ILD group and two patients (7.6%) in the ILD group (*p* = 0.29). There were no cases of acute exacerbation in the ILD group. CIRT for stage I NSCLC was as safe in the ILD group as in the non-ILD group. Coexisting ILD was a poor prognostic factor in CIRT for clinical stage I lung cancer.

## 1. Introduction

Lung cancer occurs in 7–40% of patients with interstitial lung disease (ILD); the causes are similar to those in other groups and include smoking and old age; however, fibrosis itself can be a risk factor for ILD patients [1,2,3,4].

However, anti-cancer treatments for lung cancer patients with ILD are challenging because of the risk that cancer treatments such as surgery, chemotherapy, immunotherapy, and radiotherapy can cause acute exacerbation (AE) of ILD [5,6,7,8,9,10,11,12,13,14,15,16].

Although stereotactic body radiotherapy (SBRT) using X-rays for stage I non-small cell lung cancer (NSCLC) has been reported to be safe with good outcomes [17,18,19], many prospective studies exclude cases with ILD. This is because patients with concomitant ILD have a higher risk of developing both radiation pneumonia and AE of ILD, which can be fatal [20,21,22,23,24].

At our hospital, carbon-ion radiotherapy (CIRT) has been used for clinical stage I NSCLC since 2010. Compared with X-rays, carbon ions have higher linear energy transfer and larger relative biological effectiveness (RBE) [25]. The physical characteristics of carbon ions, such as the Bragg peak and small lateral scattering, are theoretically superior to those of X-rays, allowing carbon ions to provide a more localized delivery of the radiation dose. With this benefit, CIRT for clinical stage I NSCLC has demonstrated favorable local control while minimizing damage to lung tissues [26,27,28]. However, there is little information on how the safety and efficacy of CIRT compares between patients with and without ILDs.

In the present study, we evaluated whether the presence or absence of ILD affects the clinical outcomes of CIRT for clinical stage I NSCLC.

## 2. Materials and Methods

### 2.1. Patients

This single-institution retrospective study was approved by the institutional review board of Gunma University Hospital (trial registration number: HS2019-233). Eligibility criteria for CIRT are clinically or histologically diagnosed stage I primary lung cancer, inoperable or refused surgery, and an Eastern Cooperative Oncology Group’s scale performance status between 0 and 2. Exclusion criteria include a previous history of radiotherapy near the target, an intractable infectious disease, a second active malignancy, and inability to obtain patient consent. Patients who underwent CIRT at Gunma University heavy ion medical center between June 2010 and December 2019 were included in the study. All patients were histologically or clinically diagnosed with stage I NSCLC and were confirmed as eligible for CIRT by the multidisciplinary tumor board of Gunma University Hospital. The tumor board consists of radiologists, respiratory physicians, respiratory surgeons, and radiation oncologists, and who discuss the clinical diagnosis and treatment plan. The indication of surgery was judged by a respiratory surgeon based on the respiratory function, history and complications. All patients underwent the following evaluations: whole blood cell count, differential count, routine chemistry measurements, electrocardiogram, pulmonary function tests, chest radiography, chest computed tomography (CT), abdominal CT, whole-brain magnetic resonance imaging or CT, and ^18^F-fluorodeoxyglucose positron emission tomography. In cases where either bronchoscopy or CT-guided biopsy was possible, a biopsy was performed for definitive diagnosis. If there were enlarged mediastinal lymph nodes that were difficult to distinguish from lymph node metastases on imaging, and examination was possible based on the patient’s general condition and the location of the lymph nodes, endobronchial ultrasound-guided transbronchial needle aspiration (EBUS-TBNA) was performed. To identify those patients with complications due to ILD, the pre-treatment CT images and medical history of all patients were reviewed. In this study, the primary endpoint was a difference in the incidence of adverse events according to the presence or absence of ILD. The secondary endpoints were overall survival (OS), cause-specific survival (CSS), and local control (LC).

### 2.2. Carbon-Ion Radiotherapy

CIRT was performed as described previously [27]. Patients were positioned in customized cradles (Moldcare; Alcare, Tokyo, Japan) and immobilized using thermoplastic shells (Shellfitter; Kuraray, Osaka, Japan). CT images were acquired at 2 mm slice thickness and used for the treatment planning. Contouring of the gross tumor volume (GTV) was performed on the planning CT using a lung window. The clinical target volume (CTV) was defined as the GTV plus a margin of 5 to 8 mm. The planning target volume (PTV) margin included the setup and internal margins, which were determined by assessing tumor motion on four-dimensional CT images. If the patient’s breathing rhythm was stable, respiratory gated irradiation was performed (Anzai Medical Co., Ltd., Tokyo, Japan).

Carbon-ion doses were prescribed using units of Gy (RBE), calculated by multiplying the physical dose (Gy) by the RBE values for carbon ions. The RBE values for carbon ions were obtained from a biological linear-quadratic model constructed by the Japanese National Institute for Radiological Science [29,30].

The clinical dose distribution was calculated with XiO-N treatment planning software (Elekta/Mitsubishi Electric) and was used to calculate the passive scattering carbon-ion dose distribution. From 2010 to 2015, the prescribed doses were 52.8 Gy (RBE) for T1 tumors and 60.0 Gy(RBE) for T2 tumors, delivered in four fractions, whereas since 2016, all cases were treated with 60.0 Gy(RBE) in four fractions.

### 2.3. Assessment of Interstitial Lung Disease

Two respiratory physicians (K.Y. and S.K.) reviewed the pre-treatment CT images of all patients to determine the presence of ILD. High-resolution CT (HRCT) imaging was performed using a multidetector CT scanner (Aquilion 64; Toshiba Medical Systems, Tochigi, Japan).

The ground-glass opacity (GGO) and fibrosis scores of ILD were determined according to previous reports [31,32,33]. Briefly, the percentages of GGO and fibrosis in each of the five lung lobes under three limited CT levels (the mid-aortic arch, left tracheal bifurcation, and 1 cm above the diaphragm) were evaluated, with scores of 0 to 5 given for each lobe, and the five scores were summed. The GGO score for each lobe was evaluated as follows: 0, none; 1, ≤5%; 2, 5% to <25%; 3, 25–49%; 4, 50–75%; and 5, >75% of the lobe. The fibrosis score was evaluated as follows: 0, no fibrosis; 1, interlobular septal thickening without honeycombing; 2, honeycombing <25%; 3, 25–49%; 4, 50–75%; and 5, >75% of the lobe.

CT findings of GGO, consolidation, reticulations, honeycombing, bronchiectasis, and cysts were also evaluated according to the Fleischner Society guidelines [34].

A diagnosis of ILD was made if any of the above findings were seen on HRCT. Although a histological diagnosis is normally required, the purpose of this study was not to treat ILD; therefore, a histological diagnosis was not required. Thus, the presence or absence of ILD was based on clinical judgment of imaging findings and patient symptoms.

### 2.4. Dose-Volume Histogram Evaluation

Irradiated lung volumes were evaluated as the total lung without the GTV volume. V*n* describes the fraction of the volume irradiated above *n* Gy or *n* Gy (RBE). The mean lung dose (MLD) was defined as the average dose to the whole lung. The V_5_, V_20_, V_30_, and MLD, which have been indicated to have an effect on the lungs, were investigated. Target coverage was evaluated as the percentage of the prescribed dose covering 95% of the CTV.

### 2.5. Follow-Up

The patients had a physical examination, toxicity assessments, and X-ray of the chest every month until 6 months after CIRT, while a thoracic CT scan and blood analyses were obtained every 3 months. After 6 months, the frequency was extended to once every 6 months, and the examination and consultation were continued until 5 years after the end of radiotherapy. In these follow-up consultations, ^18^F-fluorodeoxyglucose positron emission tomography (FDG-PET) was performed once a year instead of CT. Additional examinations were performed when recurrence was suspected, and the follow-up schedule was adjusted according to the treatment schedule and the patient’s general condition. Treatment toxicity was evaluated according to the Common Terminology Criteria for Adverse Events (CTCAE) version 4.0 [35].

### 2.6. Statistical Analysis

OS was defined as the time between the day of the first CIRT fraction and the day of the last follow-up or death. CSS was measured as the time from the date of initial CIRT until the date of death from the primary lung cancer or the last follow-up. Any death after disease progression was defined as death from the primary lung cancer. LC was measured from the date of initial CIRT to the date of the first local progression in the irradiated area or the date of the last follow-up. The OS, CSS, and LC rates were calculated using the Kaplan–Meier method. Differences in the OS, CSS, and LC rates between non-ILD and ILD groups were compared using the log-rank test. Differences in the numerical variables between the two groups were examined using Fisher’s exact test and the chi-square test. A *p*-value < 0.05 was considered statistically significant. All statistical analyses were performed using SPSS (version 26; SPSS Inc., Chicago, IL, USA).

## 3. Results

### 3.1. Patient Characteristics

Table 1 summarizes the patient characteristics. Among the 124 eligible patients, 26 (21%) had ILD. The pre-treatment sialylated carbohydrate antigen KL-6 (KL-6) level in the ILD group was significantly higher than that in the non-ILD group (*p* = 0.003), although there were no significant between-group differences in age, proportion of T-factor, prescribed dose, and pre-treatment respiratory functions including %VC and forced expiratory volume in one second.

### 3.2. CT Findings Associated with ILD

The CT findings of the 26 patients with ILD showed a median total GGO score of 0 (range, 0–4), and a median total fibrosis score of 2 (range, 0–10). Specific CT findings included GGO in 16 patients, reticulation in 22 patients, contraction bronchiectasis in seven patients, honeycombing in three patients, and emphysema in 15 patients. No patient showed consolidation or cysts on CT images. In the ILD group, 18 cases (66.7%) had a tumor inside the stromal shadow. CT imaging and the dose distribution in a representative case with ILD are shown in Figure 1.

### 3.3. Dose-Volume Histogram Parameters

The dose-volume histogram parameters in the non-ILD and ILD groups are shown in Table 2. The irradiated lung volumes in the non-ILD and ILD groups evaluated as the total lung without the GTV volume were 10.4% and 9.94% in V_5_; 5.58% and 5.6% in V_20_; and 3.88% and 4.09% in V_30_, respectively. Target coverage, evaluated as the percentage of the prescribed dose covering 95% of the CTV, was 99.2% in the non-ILD group and 99.4% in the ILD group. There were no significant differences in V5, V20, V30, MLD, and CTV coverage between the two groups.

### 3.4. Tumor Control and Survival

In the non-ILD group, eight patients died from primary disease and ten died from other diseases, whereas in the ILD group nine died from primary disease and two died from other diseases. The two deaths from other diseases in the ILD group were observed at 49.2 and 69.6 months post-CIRT. The OS, CSS, and LC values are shown in Figure 2. The sites of recurrence in patients who died of primary disease were as follows: in the non-ILD group, the recurrence cites were local (*n* = 3), brain (*n* = 2), mediastinal lymph node (*n* = 1), bone (*n* = 1), and pleural effusion (*n* = 1); in the ILD group, the recurrence sites were local (*n* = 4), mediastinal lymph nodes (*n* = 4), lung (*n* = 2), liver (*n* = 2), and pleural effusion (*n* = 1). The OS rates at three and five years were 83.2% and 78.5%, respectively, in the non-ILD group, and 59.7% and 44.8%, respectively, in the ILD group. The non-ILD group showed significantly better OS than the ILD group (*p* = 0.002). The CSS rates at three and five years were 90.7% and 90.7%, respectively, in the non-ILD group, and 59.7% and 52.2%, respectively, in the ILD group. The non-ILD group showed significantly better CSS than the ILD group (*p* < 0.001). The LC rates at three and five years were 90.4% and 90.4%, respectively, in the non-ILD group, and 76.0% and 76.0%, respectively, in the ILD group. There was no significant difference in LC between the two groups (*p* = 0.096).

### 3.5. Adverse Events

Adverse events are listed in Table 3. There were no adverse events of ILD related to the CIRT. No patient developed Grade 4 or 5 radiation pneumonitis in either the ILD group or the non-ILD group. The incidence of radiation pneumonitis (Grade 2 or worse) was 4.0% (5/124) overall, and 3.0% (3/98) in the non-ILD group and 7.6% (2/26) in the ILD group.

## 4. Discussion

To the best of our knowledge, the present study is the first to compare the safety and efficacy of CIRT for NSCLC between non-ILD and ILD groups. The OS and CSS rates in the ILD group were significantly worse than those in the non-ILD group, whereas the LC rate was not significantly different. Regarding lung toxicity, no patients developed AE or fatal radiation pneumonitis, and the incidence of radiation pneumonitis of Grade 2 or 3 was not significantly different between the non-ILD and ILD groups. In addition, there was no significant difference in dose-volume histogram parameters between the two groups.

Patients with NSCLC and coexisting ILD are at an increased risk of severe adverse events after SBRT. According to retrospective studies on SBRT for lung cancer, patients with ILD have a significantly higher risk of radiation pneumonitis than patients without ILD [37]. As shown in Table 4, previous incidences of radiation pneumonitis of Grade 2 or worse and Grade 3 or worse after SBRT ranged from 19.0–50.0% and 10.0–38.9%, respectively, in the ILD group, and from 1.7–14.8% and 1.0–2.6% in the non-ILD group [21,22,38,39,40,41]. For SBRT, there were clear increases in Grade 2 or worse and Grade 3 or worse pneumonia in the ILD group. The corresponding values in our study were 7.6% and 3.8%, respectively, in the ILD group, and 3.0% and 1.0%, respectively, in the non-ILD group. Therefore, our findings showed a relatively lower incidence of radiation pneumonitis in CIRT-treated NSCLC patients with ILD than in previously reported SBRT-treated NSCLC patients with ILD.

A possible reason for the lower incidence of radiation pneumonitis after CIRT was the better dose distribution. Characteristics of carbon-ion beams include the distal fall-off at the Bragg peak and less lateral scatter than X-rays, realizing a more-conformal target dose and reducing the lung tissue dose. A previous dosimetric study comparing dose-volume histogram parameters demonstrated that the values of MLD, V5, V20, and V30 were significantly lower in CIRT than in SBRT [42]. Furthermore, Chen et al. reported that the proportions of treatment-related mortality and ILD-specific toxicity were 15.6% and 25%, respectively, for SBRT, and 4.3% and 18.2%, respectively, for particle therapy including CIRT, indicating that CIRT has the potential to deliver a safer dose to organs at risk compared with SBRT [20].

Previous studies have reported inferior survival in patients with ILD after cancer treatment compared with patients without ILD and listed Table 5. In addition to primary disease progression, the occurrence of fatal treatment-related pneumonitis and AE can decrease survival. Regarding SBRT, Ueki et al. reported three year OS and LC rates of 70.8% and 77.7%, respectively, in a non-ILD group, and 53.8% and 71.4%, respectively, in an ILD group. [23] Other studies on SBRT reported 3 year OS ranging from 54–80% in non-ILD groups and 0–50% in ILD groups [38,43,44]. A similar trend was demonstrated in a surgical series, with Sekihara et al. reporting that in patients with stage I lung cancer, five year OS was significantly higher in a non-ILD group (84.6%) than in an ILD group (44.0%) [45]. The authors also reported that death from primary disease was significantly higher in the ILD group, and that CSS rates were 89.8% in the non-ILD group and 56.6% in the ILD group. In the current study, in CIRT, OS in the ILD group was worse than in the non-ILD group (three year OS, 59.7% vs. 83.2%, respectively), with most of the deaths being due to the primary cancer, which is consistent with previous reports of OS in SBRT or surgery.

The limitations to our study should be mentioned. First, making a diagnosis of ILD is sometimes difficult; indeed, there are no standardized criteria. However, to minimize diagnostic variations, the respiratory physicians re-evaluated CT images according to current guidelines [34]. Second, the study was a retrospective single-institution study that included a relatively small number of patients. In particular, it was difficult to obtain sufficient data to make a statistical analysis of adverse events. A multicenter validation of the present results, with a larger patient cohort, is warranted in the future. Third, regarding DLCO, many patients had poor respiratory function and so, in many cases, DLCO was not measured.

## 5. Conclusions

Following CIRT for NSCLC, the incidence of radiation pneumonitis in patients with ILD was not significantly different from that in patients without ILD. Coexisting ILD was a poor prognostic factor for CIRT for clinical stage I lung cancer. However, it is necessary to validate our results through prospective multicenter trials and to explore the causes of the lower lung cancer survival rate in the ILD group compared with the non-ILD group.

## Figures and Tables

**Figure 1 cancers-13-04204-f001:**
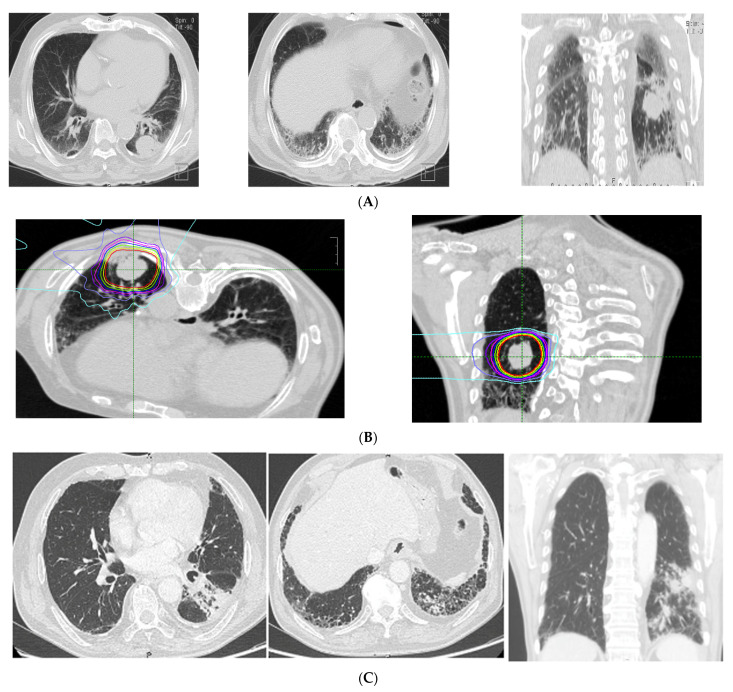
Representative case of imaging findings and dose distributions in the interstitial lung disease (ILD) group. (**A**) pre-treatment CT (diagnosis: usual interstitial pneumonitis (UIP); fibrosis score: 5, GGO score: 0, CT findings: honeycomb, GGO, reticulation, traction bronchitis, emphysema). (**B**) Dose distribution. (**C**) Post-treatment CT (6 months after CIRT).

**Figure 2 cancers-13-04204-f002:**
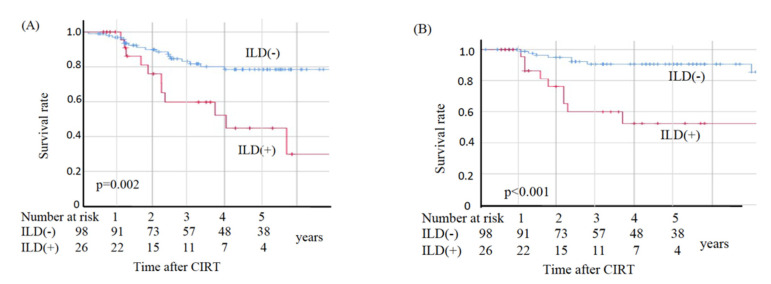
Kaplan–Meier estimates for (**A**) overall survival rates, (**B**) cause-specific survival rates, and (**C**) local control rates. Blue lines: non-ILD group (*n* = 98), red lines: ILD group (*n* = 26).

**Table 1 cancers-13-04204-t001:** Patient characteristics.

		Non-ILD Group (*n* = 98)		ILD Group (*n* = 26)		*p*-Value
Age (years)						
median (range)		74	(47–91)	76	(59–88)	*p* = 0.067
Clinical T-class (UICC 8th)						
T1		70	71.4%	18	69.2%	*p* = 0.826
T2		28	28.6%	8	30.8%	
Maximum SUV value						
median (range)		3.5	(0.8–15.6)	5.6	(2.3–20.83)	*p* = 0.009
Consolidation/Tumor ratio						
	<50%	9		0		
	50–75%	16		1		
	75%<	73		25		
Pre-treatment KL-6 level (U/mL)						
median (range)		250	(110–1507)	417	(181–923)	*p* = 0.003
Pre-treatment vital capacity (%)						
median (range)		96.2	(41.6–146.3)	88.6	(70.6–130.5)	*p* = 0.630
Pre-treatment FEV1.0 (L)						
median (range)		1.78	(0.46–4.51)	1.96	(0.93–3.73)	*p* = 0.695
Prescribed dose [Gy (RBE)]						
52.8 Gy (RBE) in four fractions		43	43.9%	6	23.1%	*p* = 0.054
60.0 Gy (RBE) in four fractions		55	56.1%	20	76.9%	
Follow-up duration (years)						
median (range)		3.4	(0.2–10.1)	3.5	(0.6–9.0)	

Abbreviations: UICC 8th, Union for International Cancer Control (8th Edition) [36]; SUV, standardized uptake value; ILD, interstitial lung disease; RBE, relative biological effectiveness.

**Table 2 cancers-13-04204-t002:** Treatment parameters.

	Non-ILD Group (*n* = 98)	ILD Group (*n* = 29)	*p*-Value
Total lung minus GTV (%) median (range)			
V5	10.4	9.94	0.908
	(3.29–23.47)	(3.12–21.15)	
V20	5.85	5.6	0.828
	(1.90–14.7)	(1.35–12.09)	
V30	3.88	4.09	0.700
	(1.29–10.96)	(0.94–9.12)	
Mean Lung Dose (cGy), median (range)			
	303.5	303.5	0.729
	(99–752)	(80–611)	
Prescribed dose covering 95% of CTV (%), median (range)			
	99.2	99.4	0.344
	(80.6–114)	(94.5–100)	

Abbreviations: Vx, Percentage of volume irradiated over X Gy (RBE); GTV, Gross Tumor Volume; CTV, Clinical Target Volume; ILD, interstitial lung disease.

**Table 3 cancers-13-04204-t003:** Adverse events (CTCAE ver.4.0).

		Non-ILD Group (*n* = 98)			ILD Group (*n* = 26)		
Radiation pneumonitis							
Grade	0	6	6.1%		1	3.8%	
	1	89	90.8%		23	88.5%	
	2	2	2.0%		1	3.8%	
	3	1	1.0%		1	3.8%	
Other than radiation pneumonitis							
Grade	0	85	86.7%		24	92.3%	
	1	3	3.1%		0	0.0%	
	2	9	9.2%	8 rib fracture, 1 chest pain, 1 hemoptysis	2	7.7%	2: rib fracture
	3	1	1.0%	pyothorax	0	0.0%	

Abbreviations: CTCAE, Common Terminology Criteria for Adverse Events.

**Table 4 cancers-13-04204-t004:** Incidence of radiation pneumonitis (Grade 2 or worse) in patients with or without interstitial pneumonitis.

Authors	Treatment Modality	Number of Patients (ILD(+)/ILD(−))	Dose and Fractionation	ILD(+)		ILD(−)	
				≥Grade 2 (%)	≥Grade 3 (%)	≥Grade 2 (%)	≥Grade 3 (%)
Yoshitake T et al. [38]	SBRT	18/242	48 Gy/4 fr	50.0	38.9	5.8	1.2
Nakamura M et al. [39]	SBRT	7/49	48–56 Gy/4 fr	28.6	NA	8.2	NA
Okubo M et al. [40]	SBRT	11/60	40–60 Gy/5–10 fr	45.5	NA	1.7	NA
Tsurugai Y et al. [41]	SBRT	42/466	40 or 50 Gy/5 fr	19.0	11.9	14.8	2.6
Glick D et al. [21]	SBRT	39/498	60 Gy/8 fr, 54–60 Gy/3 fr	20.5	10.3	5.8	1.0
Behig H et al. [22]	SBRT	30/474		NA	32.0	NA	2.0
Present study	CIRT	26/98	52.8 or 60.0 Gy (RBE)/4 fr	7.6	3.8	3.0	1.0

Abbreviations: SBRT, stereotactic body radiotherapy; CIRT, carbon ion radiotherapy; ILD, interstitial lung disease.

**Table 5 cancers-13-04204-t005:** Overall survival and local control in patients with or without interstitial pneumonitis.

Authors	Treatment Modality	Number of Patients (ILD(+)/ILD(−))	ILD(+)		ILD(−)	
			OS	LC	OS	LC
Ueki et al. [23]	SBRT	20/137	53.8% (3 y)	71.4% (3y)	70.8% (3 y)	77.7% (3 y)
Yoshitake T et al. [36]	SBRT	18/243	49.6% (2 y)	NA	86.7% (2 y)	NA
Yamaguchi S et al. [43]	SBRT	16/84	48% (3 y)	94% (3y)	54% (3 y)	88% (3 y)
Hara Y et al. [44]	SBRT	6/18	0% (3 y)	NA	67% (3 y)	NA
Sekihara et al. [45] for T1	surgery	106/1948	44.0% (5 y)	NA	84.6% (5 y)	NA
Present study	CIRT	29/98	59.7% (3 y)	76.0% (3y)	83.2% (3 y)	90.4% (3 y)

Abbreviations: SBRT, stereotactic body radiotherapy; CIRT, carbon ion radiotherapy; ILD, interstitial lung disease; OS, overall survival, LC, local control.

## Data Availability

The supporting data are not publicly available because they contain information that could compromise the privacy of the research participants.

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
