# Peer review of "Efficacy and Safety of Carbon-Ion Radiotherapy for Stage I Non-Small Cell Lung Cancer with Coexisting Interstitial Lung Disease"

_cancers, 2021, doi:10.3390/cancers13164204_

Round 1
Reviewer 1 Report
The authors took great effort to improve the style and content of the paper, and well respected (virtually) all my earlier comments. I have no further (major) suggestions.
Author Response
Thanks to your comment, I was able to improve the content. Thank you very much.
Reviewer 2 Report
The revised manuscript is nice, but further revision makes it more implicated.
Major
- Line 72; Method of patient selection for CIRT should be described.
1) How about the discussion of cancer bord?
2) How do surgeons contribute to patient selecting for CIRT? - Line 75; Inclusion criterion and exclusion criterion of CIRT should be described concretely.
Minor
- Line 39; (%) is needed.
- All stage I have to be changed to clinical stage I
- Table 1; T- class to clinical T-crass
Author Response
Response to Reviewer 2 Comments
<Major>
Point 1:
Line 72; Method of patient selection for CIRT should be described.
1) How about the discussion of cancer bord?
2) How do surgeons contribute to patient selecting for CIRT?
Response 1: Review board in Line 72 means ethics committee, so it should be revised along with the explanation of tumor board in Line 75. The following information has been added to Line 82-85.
The Tumor board consists of radiologists, respiratory physicians, respiratory surgeons, and radiation oncologists, and who discuss the clinical diagnosis and treatment plan. The indication of surgery was judged by respiratory surgeon based on the respiratory function, history and complications.
Point 2:
Line 75; Inclusion criterion and exclusion criterion of CIRT should be described concretely.
Response 2: The following information has been added to Line73-78.
Eligibility criteria for CIRT are clinically or histologically diagnosed stage I primary lung cancer, inoperable or refused surgery, and an Eastern Cooperative Oncology Group's scale performance status between 0 and 2. Exclusion criteria include a previous history of radiotherapy near the target, an intractable infectious disease, a second active malignancy, and inability to obtain patient consent.
<Minor>
Point 3:
Line 39; (%) is needed.
Response 3: We have corrected this as suggested (lines 39).
Point 4:
All stage I have to be changed to clinical stage I
Response 4: We have corrected this as suggested (lines 26,32,36,39,63,68,72,316).
Point 5:
Table 1; T- class to clinical T-crass
Response 5: We have corrected this as suggested (Table 1).

Round 2
Reviewer 2 Report
The latest version of the manuscript has been nicely revised.